# Data-driven analytics for student reviews in China's higher vocational education MOOCs: A quality improvement perspective

Hongbo Li[1], Huilin Gu[1], Xue Hao[1]*, Xin Yan[2], Qingkang Zhu[3]*

1 School of management, Shanghai University, Shanghai, China, 2 School of Cultural Heritage and Information Management, Shanghai University, Shanghai, China, 3 Department of APP Research and Development, Baidu Inc., Shanghai, China

* 15264459383@163.com (XH); zqkang1112@qq.com (QZ)

**Data Availability Statement:** Publicly available datasets were analyzed in this study. This data can be found here: https://github.com/XueHao-23/MOOC-reviews.

## Abstract

Higher vocational education is the core component of China's national education system and shoulders the mission of cultivating high-skilled and applied talents. The wide application of Massive Open Online Courses (MOOCs) has effectively improved the curriculum system of China's higher vocational education. In the meantime, some MOOCs suffer from poor course quality. Therefore, from the perspective of sustainable course quality improvement, we propose a data-driven framework for mining and analyzing student reviews in China's higher vocational education MOOCs. In our framework, we first mine multi-level student demands hidden in MOOC reviews by combining web crawlers and text mining. Then we use an artificial neural network and the KANO model to classify the extracted student demands, thereby designing effective and sustainable MOOC quality improvement strategies. Based on the real data from China's higher vocational education MOOCs, we validate the effectiveness of the proposed data-driven framework.

## 1. Introduction

As the core component of China's national education system, higher vocational education undertakes the mission of cultivating high-skilled and applied talents, effectively promoting the sustainable development of society and economy. With the development of education informatization and digitalization, Massive Open Online Courses (MOOCs) have been widely adopted in the teaching of higher vocational education due to their advantages such as flexibility of time and place, breaking the area boundary, and realizing the goal of sharing resources [1]. As of 2020, China has built more than 220,000 vocational education online courses, but the quality of many MOOCs is still low [2]. Therefore, how to effectively improve the quality of MOOCs is a key issue that needs to be solved in the process of the sustainable development of higher vocational education.

It is an important means to improve the quality of MOOCs by understanding and satisfying students' differentiated demands for MOOC teaching, and then improving students' satisfaction with the courses [3]. The degree of meeting student demands affects student satisfaction

**Funding:** This research was funded by the Soft Science Project of Shanghai Science and Technology Innovation Action Plan (Grant Number 23692113000).

**Competing interests:** The authors have declared that no competing interests exist.

with MOOCs. In this study, we use the positive and negative sentiment polarity weights of demands to measure the influence of demand on student satisfaction and use the KANO model to explain the relationship between the degree of meeting various demands and student satisfaction with MOOCs. There have been many studies on student demands for online course learning, and these studies mainly focus on two perspectives, i.e., student demand acquisition and classification. Most of these studies dissected students' demands through user interviews [4] and literature analysis [5], which suffer from problems such as subjective flaws and high information acquisition costs.

The resources in MOOCs basically are in the form of digitization. If students' various demands for courses can be directly extracted from MOOC resources, it will effectively improve MOOCs. The development of artificial intelligence technologies, such as data mining and machine learning, has provided effective tools for the mining and analysis of MOOC data. In recent years, data mining for MOOCs has attracted the attention of scholars. For example, many scholars have explored the forum data on MOOC websites [6]. However, few scholars have paid attention to the MOOC reviews to identify students' demands and improve the course quality. Typically, forums on MOOC websites support student interaction to learn course content, ask questions, discuss specific topics, and share opinions and experiences, covering a wide range of topics. In this study, MOOC reviews are student evaluations of MOOC resources, which are biased towards students' experiences of using them. MOOC reviews can be seen as part of forum data on MOOC websites. In addition, most of the existing studies also overlook higher vocational education MOOCs as research objects.

In general, the existing studies not only rarely explore students' differentiated demands implied in the MOOC reviews but also lack attention to MOOCs in higher vocational education. Therefore, from the perspective of course quality improvement, we investigate the mining of higher vocational education MOOC reviews and propose a data-driven framework for mining and analyzing student reviews in China's higher vocational education MOOCs. In our framework, we first mine multi-level student demands hidden in MOOC reviews by combining web crawlers and text mining. Then we use an artificial neural network and the KANO model to classify the extracted student demands, thereby designing effective and sustainable MOOC quality improvement strategies. Our framework not only fills the gap in the field of MOOC review studies in higher vocational education using data mining techniques but also provides a theoretical and practical basis for curriculum design, teaching method improvement, and platform optimization of higher vocational education MOOCs. Based on the real data from China's higher vocational education MOOCs, we validate the effectiveness of the proposed data-driven framework and provide constructive suggestions for educational institutions, course designers, and online education platform developers to improve the quality of courses, enhance student satisfaction, and promote the development of higher vocational education.

The remainder of this paper is organized as follows. Section 2 gives a literature review on the topics related to this paper. The data-driven framework for mining and analyzing MOOC reviews in China's higher vocational education is presented in Section 3. Sections 4 and 5 discuss the two parts of our framework in detail. In Section 6 and Section 7, we conduct computational experiments to validate the effectiveness of our framework. The last section concludes the paper and discusses future research directions.

## 2. Literature review

### 2.1. MOOCs quality improvement

Existing studies on the methods of online course quality improvement mainly focus on the perspective of learners. Hsieh [7] investigates strategies to improve the quality of MOOCs by

analyzing students' learning demands for MOOCs in Taiwan. Luo and Ye [8] propose a holistic quality criteria framework to investigate the quality criteria of language MOOCs from the learners' perspective. Bigatel and Edel-Malizia [9] use the indicators of engaged learning online framework to evaluate the quality of online courses. Stracke [10] introduces the quality reference framework for the quality of MOOCs. Albelbisi [11] develop and validate the MOOC success scale in the Malaysian context, and derive six factors related to MOOC success. Lowenthal and Hodges [12] investigate the quality of MOOCs by using the quality matters quality control framework. Su et al. [13] construct a quality evaluation index system for MOOCs based on the user perspective. Ferreira [14] propose a quality criteria in MOOC as a mechanism for the university to approve or disapprove a MOOC (checklist) and assess its quality. Samed Al-Adwan & Khdour [15] investigate both the cognitive and psychological influential factors that determine the readiness of Jordanian students to adopt MOOCs, and help higher education institutions in the enhanced design of MOOCs. Zhang et al. [16] study the learning behaviors of dropouts in MOOCs, and propose suggestions for MOOC design to prevent dropout.

## 2.2. Student demands in online courses

The existing literature on online course student demands analysis can be divided into two categories according to the research methods: (a) analyzing student demands using classical methods such as interviews; and (b) analyzing student demands using data mining.

**2.2.1 Traditional methods for student demands analysis.** Researchers have analyzed student demands in the online course environment from two perspectives:

(a) The perspective of student demand acquisition. Rezaie et al. [4] identify students' demands in Iran's virtual higher educational systems with experimental surveys and personal interviews. Liu et al. [17] obtain the student demands for online learning using questionnaire surveys. Chaveesuk et al. [18] investigate the key factors that influence behavioral intention to adopt MOOCs using a structured questionnaire. Moore and Blackmon [5] report the results of a systematic review of learners' experiences and perspectives in MOOCs to identify learners' demands. Eom and Ashill [19] study university students' demands for online courses based on the constructivist learning theory. It should be noted that the student demands extracted through interviews and literature analysis have some limitations, such as strong subjectivity.

(b) The perspective of student demands classification. The KANO model [20] is usually used to identify students' multi-level demands. Dominici and Palumbo [21] study students' must-be and one-dimensional demands for online courses based on questionnaire data. Bauk [22] assesses students' various demands in the context of online learning. The KANO model is a qualitative analysis method that cannot accurately classify each demand, and the cost of quantifying the KANO model by questionnaire surveys is relatively high. These pose some challenges to the effective analysis of student demands for online courses.

**2.2.2 Data mining methods for student demands analysis.** With the development of artificial intelligence technologies, researchers have started to focus on MOOC data mining from multiple perspectives, such as visual analysis and learning effect evaluation. Fu et al. [6] present a design study for developing an interactive visual analytics system based on MOOC forum data. Chiu and Hew [23] investigate the factors influencing peer learning and performance in MOOC asynchronous online discussion forums. Wei et al. [24] propose a deep neural network model for MOOC forum post classification. In addition, a few scholars have focused on the MOOC review data. Geng et al. [25] explore the focal points and sentiments of learners in MOOC reviews. Hew et al. [26] analyze the relationship between students' implicit demands and satisfaction in MOOC reviews. Yang [27] uses word frequency and co-

occurrence analysis, comparative keyword analysis, and structural topic modeling to analyze reviews from one Massive Online Open Courses (MOOCs) platform in China.

In summary, existing research has made significant progress in the areas of MOOC quality improvement and student demand analysis. In terms of course quality improvement, researchers have developed a variety of models and frameworks from the learner's perspective. In terms of student demand analysis, traditional methods such as interviews and questionnaire research, as well as demand categorization based on the KANO model, while providing valuable perspectives for understanding students' demands, have also revealed limitations in subjectivity and cost. Studies based on MOOC data mining are showing potential for uncovering students' implicit demands and improving course quality. Although these studies provide important theoretical and practical foundations, there is still a noticeable gap in the field of MOOCs in higher vocational education, especially in the mining of differentiated student demands in MOOC review data. Existing studies tend to overlook the importance of extracting specific demands from MOOC student reviews. Therefore, this study aims to fill this gap. We propose a data-driven framework to analyze MOOC review data in higher vocational education to extract specific demands and design course quality improvement strategies accordingly. The goal of this study is not only to improve the understanding of student demands for MOOCs but also to provide practical insights and strategies for improving the quality of higher vocational education MOOCs

## 3. A data-driven framework for mining and analyzing MOOC reviews in higher vocational education

In order to effectively improve the quality of higher vocational education MOOCs, we propose a data-driven framework for mining and analyzing MOOC reviews in higher vocational education (Fig 1). The framework consists of the following two parts:

(a) Mining student demands from MOOC reviews. Web crawlers and text mining are adopted to extract useful information from MOOC reviews. Specifically, first, web crawlers are designed to automatically collect data from MOOC websites. The data contains textual reviews and ratings given by students. Various pre-processing techniques are applied to the unstructured reviews: splitting reviews, deleting extremely short reviews, cleaning the text and tagging the part-of-speech of words. This results in the processed reviews data $r_c$. $r_c$ and the structured rating data $y_c$ form the initial dataset $D_1$. Second, the dataset $D_1$ is analyzed using text mining algorithms. Topics are extracted from $D_1$ using the latent Dirichlet allocation (LDA) method. Word vectors are trained using the Word2vec method and the words that are similar to each high-frequency word in each topic are found. Then the student demands can be identified. Third, using dataset $D_1$ to construct the sentiment lexicon that is used to perform sentiment

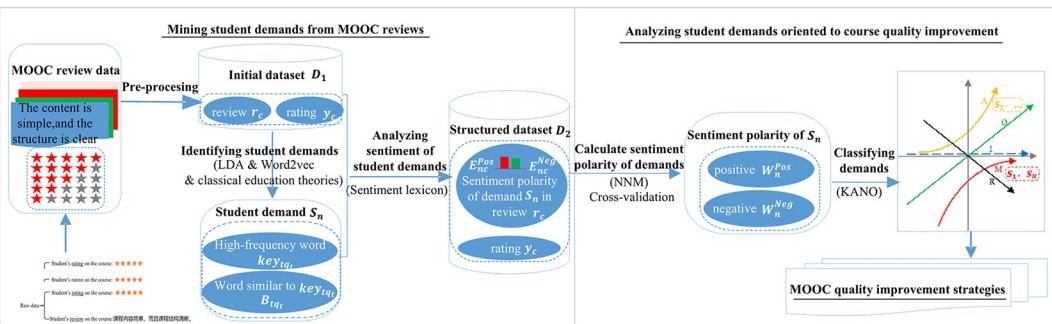

**Fig 1. Data-driven framework for mining and analyzing MOOC reviews.**

analysis on the previously extracted students' demands. In this way, we can determine the sentiment polarity of each demand implied in each review and the resulting structured data set is labeled as $D_2$.

(b) Analyzing students' demands oriented to sustainable course quality improvement. The previous structured dataset $D_2$ is analyzed from multiple perspectives using an artificial neural network and KANO model. We aim to obtain the category of each demand and the MOOC quality improvement strategies. Specifically, we first use K-fold cross-validation method to configure the parameters of the Neural Networks Model (NNM) that is used to measure the impact of student demand sentiment on the satisfaction (rating) in MOOCs. Then for each demand, we obtain the positive and negative sentiment weights that are employed to design a quantitative demand classification method based on the KANO model. Finally, the category of each demand can be identified and course quality improvement strategies for the MOOCs can be developed based on the importance of each demand. In the next two sections, we will describe the above two parts of our framework in detail.

## 4. Mining student demands from MOOC reviews

### 4.1. MOOC review data collection and processing

We designed web crawlers to collect MOOC review data. The raw data exists on the MOOC websites, and Fig 2 provides an example of the raw data on MOOC websites. For each course, the crawled data includes the ratings and textual reviews of all students. The ratings of MOOCs have 5 levels, where 1-star indicates extreme dissatisfaction and 5-star indicates extreme satisfaction.

Then the crawled raw data is processed in the following steps:

- Extremely short reviews that contain less than 5 Chinese characters are deleted, because they usually contain meaningless or low-information words.

- Each review is split into sentences based on punctuation marks to facilitate the extraction of phrases containing student demands.

- The text is cleaned by deleting numbers, punctuation marks and non-Chinese characters.

- Stop words (such as "而且"("and"), "否则"("otherwise"), etc.) are removed.

- Chinese text (a sequence of Chinese characters) is split into words.

- The part-of-speech of words is tagged.

The processed initial dataset is denoted as $D_1 = \{(r_c, y_c)\}_{c=1}^{C}$, where $C$ is the total number of reviews in $D_1$, $r_c$ indicates the data obtained after processing the $c$-th review, and $y_c$ denotes the corresponding student satisfaction (rating), $c = 1, 2, \ldots, C$. $r_c = \{r_{c1}, r_{c2}, \ldots, r_{cM_c}\}$, where $r_{cm_c}$ denotes the data obtained after processing the $m_c$-th phrase of the $c$-th review, and it consists of the words and their part-of-speech. $M_c$ denotes the total number of phrases in the $c$-th

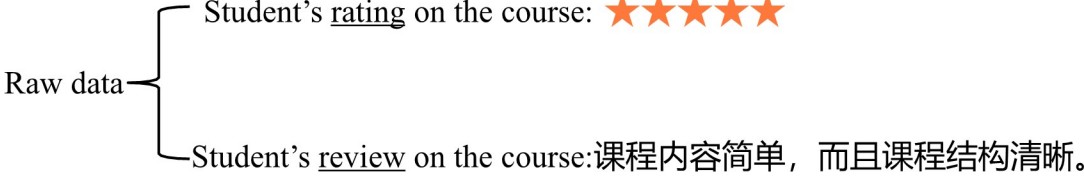

**Fig 2. An example of ratings and reviews on a MOOC website. (for illustrative purpose only).**

review, $m_c = 1,2,\ldots,M_c$. For example, the $c$-th raw review "课程内容简单，而且课程结构清晰。" ("The course content is simple and the course structure is clear.") is processed as follows: $R_c = \{r_{c1}, r_{c2}\} = \{$"课程内容/名词、简单/形容词", "课程结构/名词、清晰/形容词"$\}$ ($\{$"Course content/noun, simple/adjective", "Course structure/noun, clear/adjective"$\}$).

## 4.2. Identifying student demands for MOOCs

First, we obtain the latent topics in the reviews based on LDA. LDA is a topic model that can output the latent topics described by some high-frequency words [28]. The student demands in MOOC reviews are usually nouns or noun phrases [29]. We input the nouns (phrases) of each review in dataset $D_1$ into the LDA model to mine the latent topics. We also remove the noise words in these topics and merge similar topics. Let $T$ represent the number of topics extracted from dataset $D_1$ by LDA. Then the $t$-th topic $B_t = \{key_{t1}, key_{t2}, \ldots, key_{tQ_t}\}(t = 1, 2, \ldots, T)$, where $key_{tq_t}$ is the $q_t$-th high-frequency word in the $t$-th topic and $Q_t$ denotes the number of high-frequency words in $B_t$, $q_t = 1,2,\ldots,Q_t$. We assume that when the cosine similarity of any two words is greater than a pre-defined value $\beta$, we say that these two words are similar. The nouns (phrases) similar to $key_{tq_t}$ can be represented as $B_{tq_t} = \{term^1_{tq_t}, term^2_{tq_t}, \ldots, term^Z_{tq_t}\}$, where $term^z_{tq_t}$ denotes the $z$-th word similar to $key_{tq_t}$, $z = 1, 2, \ldots, Z$.

Then, we get student demands by finding similar words to the high-frequency words using Word2vec. The Word2vec model maps each word into a real-valued vector and determines the similarity among words based on the cosine similarity [30]. The word vectors are trained using dataset $D_1$ by the Word2vec method and used to find $term^z_{tq_t}$ similar to $key_{tq_t}$ to obtain $B_{tq_t}$. Let $S_n$ denote the $n$-th demand, where $n = 1,2,\ldots,N$ and $N$ is the number of student demands identified from dataset $D_1$. If there are no words similar to $key_{tq_t}$ in dataset $D_1$ and the value of $Q_t$ is small ($Q_t<10$), the topic is deleted ($B_t$ is not representative); Otherwise, let $S_n = B_t \cup B_{t1} \ldots \cup B_{tq_t} \ldots \cup B_{tQ_t} = \{word_{n1}, word_{n2}, \ldots, word_{nF_n}\}$, where $word_{nf_n}$ denotes the $f_n$-th word of the $n$-th demand, $F_n$ is the number of words in $S_n$, and $f_n = 1,2,\ldots,F_n$.

Finally, a label is assigned to each $S_n$ according to classical education theories [31] to give the demand a practical meaning in the context of MOOCs. It means that the characteristics of MOOCs in higher vocational education are effectively analyzed according to classical education theories. And the labels for each demand $S_n$ are inferred from these characteristics, including teacher, content, structure, improvement, etc.

We use an example to explain the main points in the above approach. Table 1 shows an example of the n-th demand for MOOCs. In this example, the $t$-th topic $B_t =$ {课程大纲, 课程进 , 课程体系} ({course outline, course progress, course system}) is output by the LDA model. Then we can find the similar words for the high-frequency words in $B_t$ through the Word2vec model. For example, the words similar to "course outline", "course

**Table 1. Example of the $n$-th demand for MOOCs.**

| $B_t$ | $B_{t1} \ldots \cup B_{tq_t} \ldots \cup B_{tQ_t}$ | $S_n = B_t \cup B_{t1} \ldots \cup B_{tq_t} \ldots \cup B_{tQ_t}$ |
|---|---|---|
| 课程大纲 (course outline) | 课程层次,课程框架 (course level, course framework) | 课程大纲,课程进度,课程体系,课程层次,课程框架,课程进度条,课程计划,课程章节,课程环节,课程组织 |
| 课程进度 (course progress) | 课程进度条,课程计划 (course progress bar, course plan) | (course outline, course progress, course system, course level, course framework, course progress bar, course plan. course chapter, course section, course organization) |
| 课程体系 (course system) | 课程章节,课程环节,课程组织 (course chapter, course section, course organization) | |

progress " and "course system" can be represented as $B_{t1} = \{$课程层 ，课程框架$\}$ ({course level, course framework}), $B_{t2} = \{$课程进度条，程计划$\}$ ({course progress bar, course plan}), and $B_{t3} = \{$课 章节，课程环节，课程组织$\}$ ({course chapter, course section, course organization}) respectively. Therefore, $S_n = B_t \cup B_{t1} \cup B_{t2} \cup B_{t3} = \{$course outline, course progress,. . ., course organization$\}$. $S_n$ can be explained as "(course) structure" and the $n$-th demand is "(course) structure ".

### 4.3. Determining the sentiment polarity of student demands

Based on the previously identified student demands, we construct a sentiment lexicon by manually marking the sentiment polarity (positive or negative) of adjectives in dataset $D_1$. This lexicon is used to determine the sentiment polarity of demands in each review [32]. Specifically, we extract sentences ($r_{cm_c}$,. . .) containing student demands (words in $S_n$) from each review in dataset $D_1$. The sentiment polarity of each demand in each review is obtained by locating the demand and its adjacent sentiment words in each review phrase. The sentiment polarity of the demand is determined according to the polarity of the sentiment words in the sentiment lexicon.

As shown in Table 2, the $c$-th review can be expressed as $x_c = (E_{1c}^{Pos}, E_{1c}^{Neg}, E_{2c}^{Pos}, E_{2c}^{Neg}, \ldots, E_{NC}^{Pos}, E_{NC}^{Neg})$. In $x_c$, if the sentiment polarity of $S_n$ is positive, let $E_{nc}^{Pos} = 1$ and $E_{nc}^{Neg} = 0$; If the sentiment polarity of $S_n$ is negative, let $E_{nc}^{Pos} = 0$ and $E_{nc}^{Neg} = 1$; In the other cases, let $E_{nc}^{Pos} = 0$ and $E_{nc}^{Neg} = 0$. Therefore, the structured data of all MOOC reviews in higher vocational education and their rating data form the structured dataset $D_2 = \{(x_c, y_c)\}_{c=1}^C$.

## 5. Course quality improvement-oriented student demands analysis

### 5.1. Measuring the influence of demand sentiment on student satisfaction

The dataset $D_2$ constructed in the previous section stores the positive and negative sentiment polarities of student demands and student satisfaction (ratings). We train a neural network [33, 34] based on $D_2$. According to the obtained neural network, we use $W_n^{Pos}$ ($W_n^{Neg}$) to denote the positive (negative) sentiment weights of $S_n$. $W_n^{Pos}$ ($W_n^{Neg}$) measures the influence of demand sentiment on student satisfaction and $W_n^{Pos}$ is calculated as follows:

$$W_n^{Pos} = \frac{\sum_{h=1}^H w_{nh}^{Pos} \times w_h}{\sum_{n=1}^N \sum_{h=1}^H w_{nh}^{Pos} \times w_h + \sum_{n=1}^N \sum_{h=1}^H w_{nh}^{Neg} \times w_h} \tag{1}$$

where $w_{nh}^{Pos}$ ($w_{nh}^{Neg}$) represents the connection weight between the $n$-th neuron $E_n^{Pos}$ ($E_n^{Neg}$) in the input layer and the $h$-th neuron in the hidden layer in the neural network; $w_h$ is the connection

**Table 2. Structured data of MOOC reviews in higher vocational education.**

| Review | $S_1$ | | $S_2$ | | . . . | $S_N$ | |
|---|---|---|---|---|---|---|---|
| | $E_1^{Pos}$ | $E_1^{Neg}$ | $E_2^{Pos}$ | $E_2^{Neg}$ | . . . | $E_N^{Pos}$ | $E_N^{Neg}$ |
| $r_1$ | 1 | 0 | 0 | 1 | . . . | 0 | 0 |
| $r_2$ | 0 | 1 | 0 | 0 | . . . | 1 | 0 |
| . . . | . . . | . . . | . . . | . . . | . . . | . . . | . . . |
| $r_C$ | 0 | 0 | 1 | 0 | . . . | 0 | 1 |

weight between the $h$-th neuron in the hidden layer and the neuron in the output layer, $h = 1,2,\ldots,H$. The larger the value of $W_n^{Pos}$, the greater the positive preference of students to $S_n$. Similarly, $W_n^{Neg}$ is calculated as follows:

$$W_n^{Neg} = \frac{\sum_{h=1}^{H} w_{nh}^{Neg} \times w_h}{\sum_{n=1}^{N} \sum_{h=1}^{H} w_{nh}^{Pos} \times w_h + \sum_{n=1}^{N} \sum_{h=1}^{H} w_{nh}^{Neg} \times w_h} \qquad (2)$$

## 5.2. Classifying student demands and designing MOOC quality improvement strategies

We design a quantitative demand classification method that applies the KANO model [20] to identify the categories of each previously mined demand. A reasonable classification of student demands helps meet these demands and improve the quality of courses in a targeted manner.

We use the KANO model to explain the relationship between the degree of meeting student demands and the student satisfaction with MOOCs, and classify student demands into five categories (Fig 3), i.e., attractive demands (A), one-dimensional demands (O), indifferent demands (I), must-be demands (M), and reverse demands (R). The horizontal axis in Fig 3 represents the degree of meeting student demands in MOOC learning and the vertical axis represents the student satisfaction with MOOCs. In Fig 3, each curve corresponds to one of the five demand categories.

In our KANO model-based demand classification method, when the demand $S_n$ is satisfied, the student satisfaction $u_n = W_n^{Pos} \times \bar{E}_n^{Pos} + W_n^{Neg} \times \bar{E}_n^{Neg}$, where $W_n^{Pos}$ and $W_n^{Neg}$ represent the positive and negative sentiment weights of $S_n$ obtained in Section 5.1, respectively; $\bar{E}_n^{Pos}$ and $\bar{E}_n^{Neg}$ represent the sentiment polarity of $S_n$. If the sentiment polarity of $S_n$ is positive ($W_n^{Pos}$ is not 0), then $\bar{E}_n^{Pos} = 1$. If the sentiment polarity of $S_n$ is negative ($W_n^{Neg}$ is not 0), then $\bar{E}_n^{Neg} = 1$. Otherwise, $\bar{E}_n^{Pos} = \bar{E}_n^{Neg} = 0$ (both $W_n^{Pos}$ and $W_n^{Neg}$ are 0). The larger $u_n$ is, the higher the student

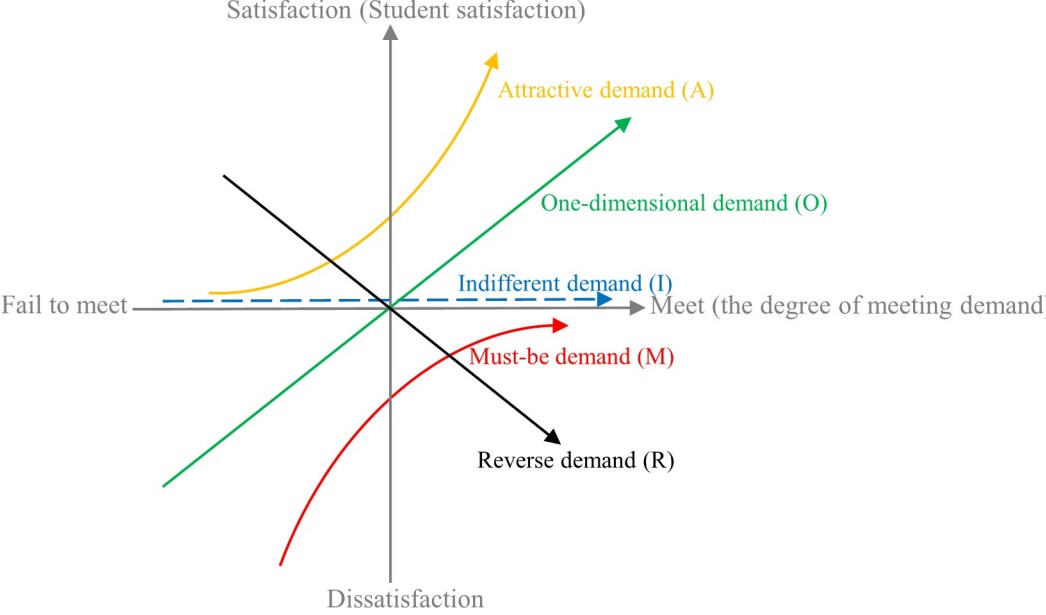

**Fig 3. The KANO model for MOOCs in higher vocational education.**

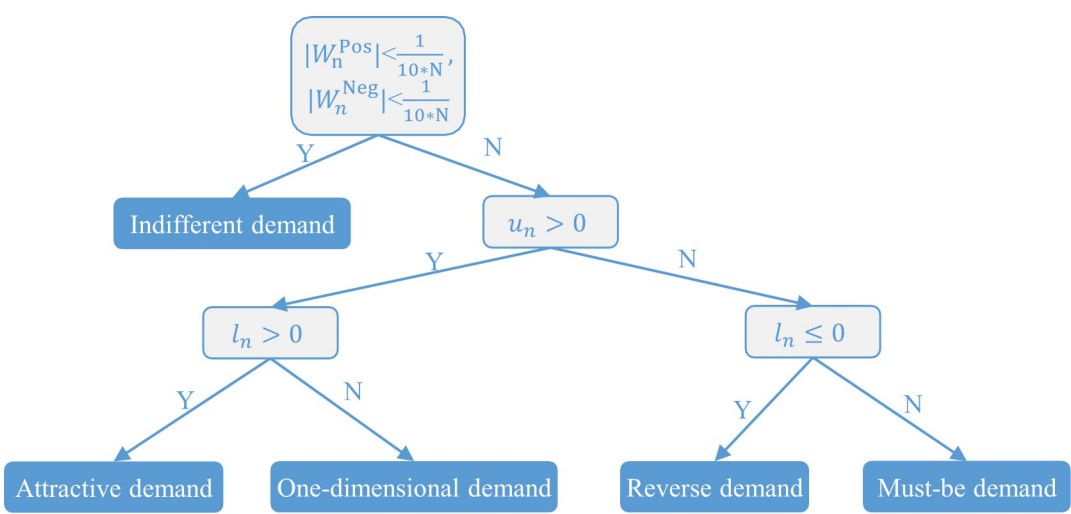

**Fig 4. Student demand classification method based on the KANO model.**

satisfaction when $S_n$ is met. In addition, the relative distance $l_n$ of the positive and negative sentiment weights of $S_n$ is calculated as $l_n = W_n^{Pos} - W_n^{Neg}$. The larger the value of $l_n$, the greater the relative affective preference of students for $S_n$.

According to $u_n$, $l_n$ and Fig 3, our student demand classification method is shown in Fig 4. Based on the classification results for the demands and the priority of each demand category in the KANO model (must-be demand > one-dimensional demand > attractive demand > indifferent demand (avoiding reverse demand)) [35], the strategies for MOOC quality improvement in higher vocational education can be designed. This will be further discussed and applied in the next section.

## 6. Experiments

### 6.1. Experimental settings

Our method has been implemented in a Python 3.6. The machine learning algorithms in the paper are from the Genism and Keras libraries. Our data is crawled from 39 courses of higher vocational education on the website of China University MOOC (www.icourse163.org). Our data consists of 40906 reviews that are published between February 3, 2018 and May 2, 2021. The data can be found here: https://github.com/XueHao-23/MOOC-reviews. The descriptive statistics of the review data are shown in Table 3 and we can see that, students are generally satisfied with these courses and the reviews on the courses are dominated by short texts.

The number of topics in the LDA model is set to 15, and the number of iterations is 1000. The word vector dimension in the Word2vec model is set to 100, and the Skip-gram model is used to train the word vectors. In addition, the cosine similarity threshold is 0.5. The parameters of the neural network with one hidden layer are configured with 10-fold cross-validation.

**Table 3. Descriptive statistics of the crawled MOOC review data.**

| Number of MOOCs | Average rating | Average length of the reviews | Number of reviews | | |
|---|---|---|---|---|---|
| | | | ($L$: the length of the review) | | |
| | | | $L<50$ | $50 \leq L < 200$ | $L \geq 200$ |
| 39 | 4.8724 | 17.5509 | 38370 | 2256 | 280 |

According to the cross-validation results, the neural network parameters are set as follows: the initial learning rate is 0.001, the number of neurons in the hidden layer is 16, and the number of cycles is 50.

## 6.2. Experimental results

**(1) Results and analysis of the student demand identification.** We identify 10 student demands from our data (Table 4). For each identified demand $S$, Table 4 shows the corresponding label, description, word example, total number of words, and frequency of occurrence in all MOOC reviews.

The relationship map of student demands based on Table 4 is displayed in Fig 5. It can be found from Fig 5 that, "teacher" and "content" are the two demands that students pay most attention to. We can infer that students are more inclined to focus on the teaching quality of teachers and the practicality of course content. The identification of these demands indicates that improvements in course design and teacher training in MOOCs are necessary. In addition, students are also concerned with "improvement", "interaction" and "environment". MOOCs should provide more interactive learning opportunities and create a better-quality learning environment.

**(2) Results and analysis of the student demand sentiment polarity.** The sentiment polarity of each demand was identified by the student demand sentiment analysis method in Section 4.3. There are 303 positive and 104 negative sentiment words in our constructed sentiment lexicon. Based on Table 4 and the sentiment lexicon, the statistical results of the sentiment polarity of each demand in the MOOC reviews are shown in Fig 6. It can be seen that the number of positive sentiment reviews for all demands is greater than the negative ones, indicating that each demand is satisfied to a high degree. Additionally, the negative sentiment reviews indicate that MOOCs leave a lot to be desired. Especially in the two core demands of

**Table 4. Student demands extracted from MOOC review data.**

| $s_n$ | Student demand (label) | Description | Example | #. of words | Frequency |
|-------|------------------------|-------------|---------|-------------|-----------|
| $S_1$ | Teacher | Teaching styles, personal characteristics, etc. | 风格, 教态 (style, teaching demeanor) | 68 | 24279 |
| $S_2$ | Content | Course content, learning objectives, etc. | 互联网, 物联网 (Internet, Internet of Things) | 96 | 17964 |
| $S_3$ | Structure | Course structure and organization, etc. | 层次, 体系 (hierarchy, system) | 21 | 1210 |
| $S_4$ | Improvement | Improvement of students' abilities | 能力, 眼界 (ability, perspective) | 48 | 2862 |
| $S_5$ | interaction | Interaction with teachers and students | 互动, 合作 (interaction, collaboration) | 15 | 4036 |
| $S_6$ | Assessment | Assessment of learning progress | 测试题, 考试 (test, exam) | 29 | 1487 |
| $S_7$ | Time | Time required for course study | 课余, 碎片化 (after-class, fragmented) | 13 | 820 |
| $S_8$ | Difficulty | Course difficulty and target audience | 难度, 入门 (difficulty, beginner level) | 14 | 956 |
| $S_9$ | Environment | Requirements for the learning environment | 件, 网站 (application, website) | 25 | 2491 |
| $S_{10}$ | material | Materials required for learning | 课件、视频 (courseware, video) | 15 | 1774 |

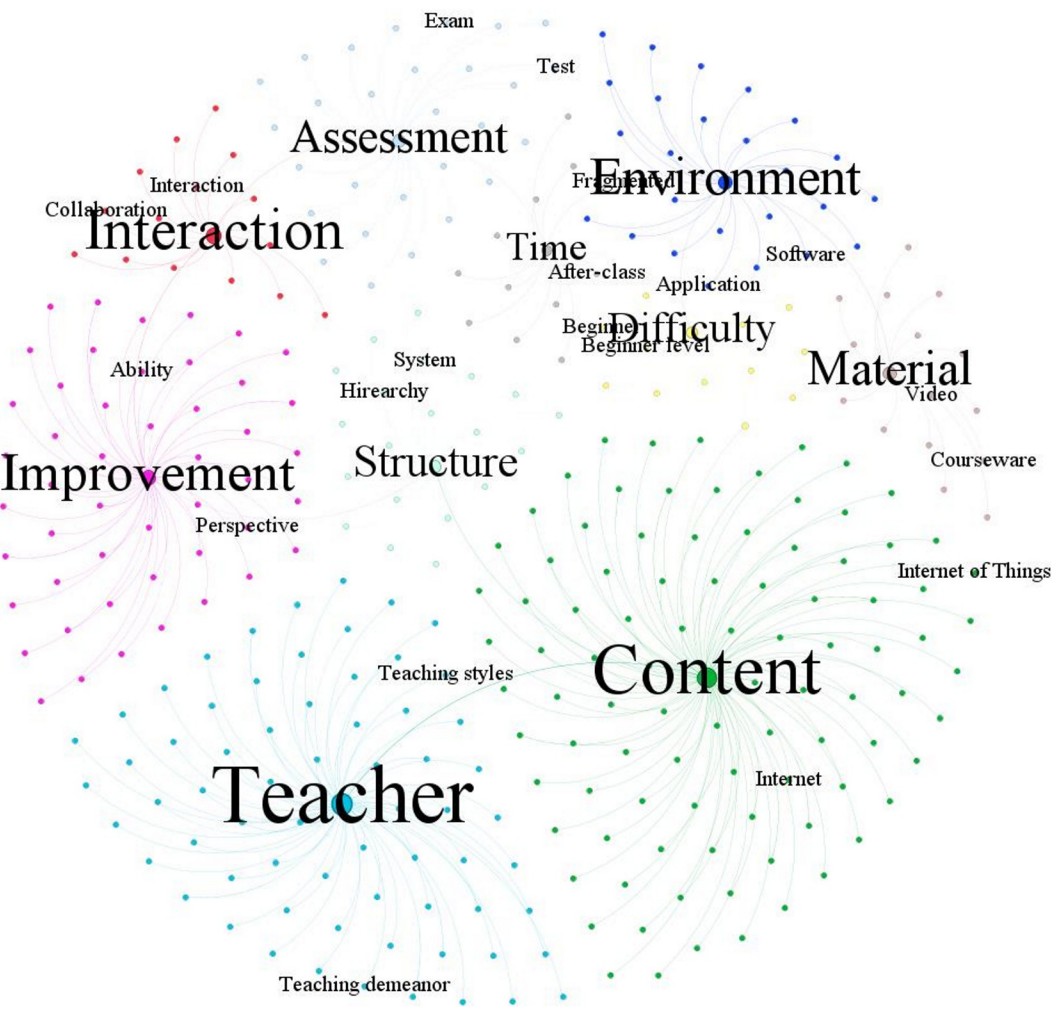

**Fig 5. Relationship map of student demands.**

"teachers" and "content", the high proportion of positive sentiment indicates that these aspects are highly satisfied, but the negative feedback should not be ignored, which provides important clues for subsequent course improvement.

(3) **Results and analysis of the sentiment polarity weights of student demands.** The loss and accuracy as the training of the neural network are shown in Figs 7 and 8. It can be seen from Figs 7 and 8 that, when the number of iterations is 30, the performance of the neural network tends to be stable, thus obtaining the final neural network. In this case, the loss of the neural network is 0.1466 and the accuracy is 0.9434, which indicates that the neural network can fit the data well and is suitable for modeling the relationship between the sentiment of the student demands and students' satisfaction. According to Formulas (1) and (2), the calculation results of $W_n^{Pos}$ and $W_n^{Neg}$ are shown in Table 5. The $W_n^{Pos}$ and $W_n^{Neg}$ of $S_n$ measure the influence of demand sentiment on student satisfaction.

(4) **Results and analysis of the student demand classification.** Using the student demand classification method shown in Fig 4, the classification results of student demands are shown in Fig 9. The student demands are classified into three categories:

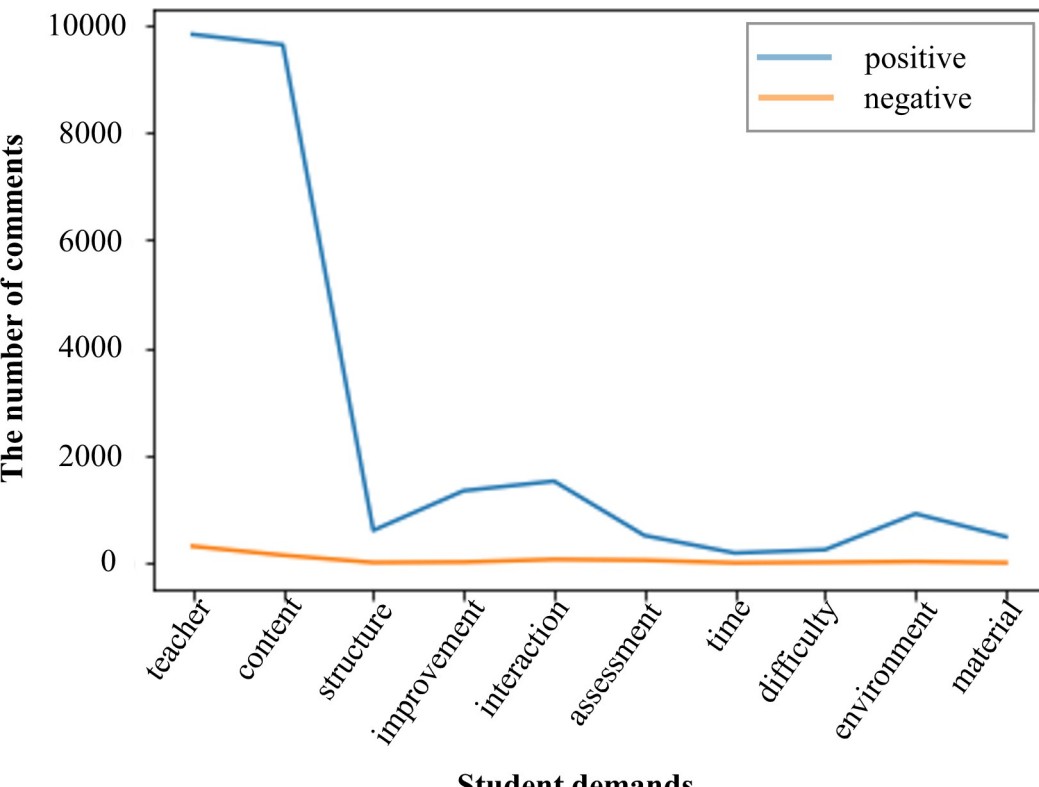

**Fig 6. The sentiment polarity of student demands in MOOC reviews.**

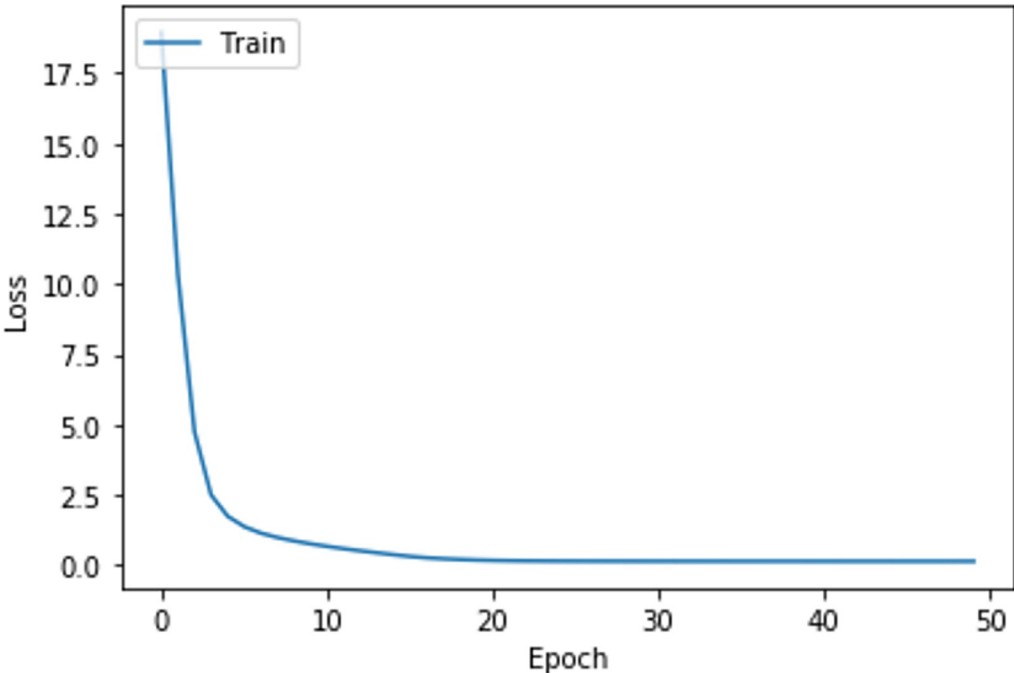

**Fig 7. The loss as the training of the neural network.**

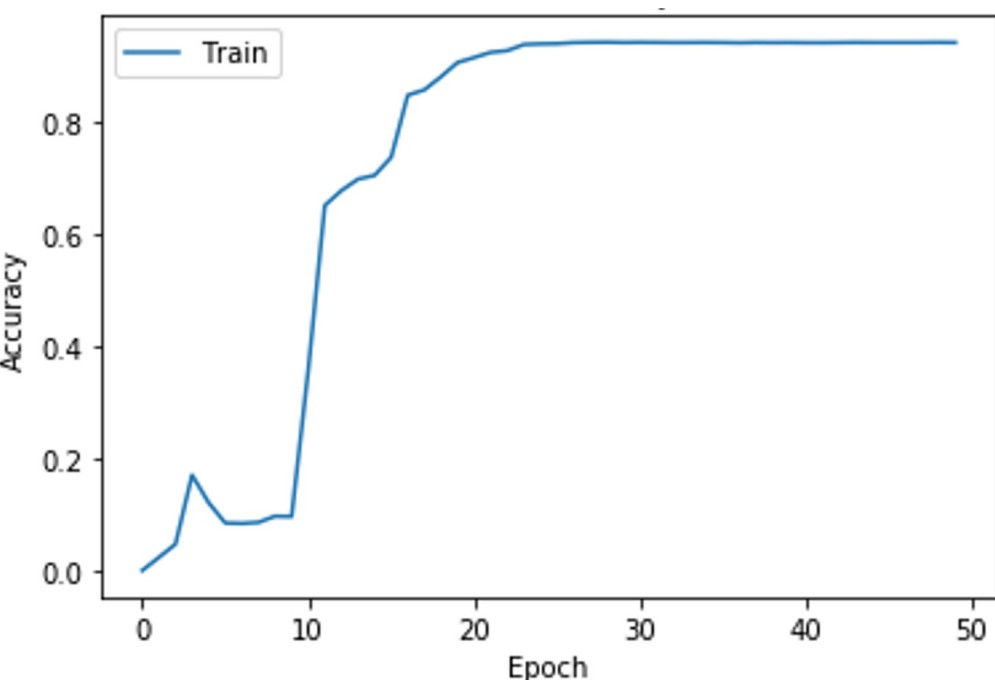

**Fig 8. The accuracy as the training of the neural network.**

- Must-be demands (M): this type of demands is related to "teacher, assessment, time". They are the basic demands of students. And when not met, students can be extremely dissatisfied.

- One-dimensional demands (O): this type of demands is related to "content, structure, inter-action, difficulty, material". These demands are some of the additional features that students expect to see in MOOCs. There is a linear relationship between the degree of meeting this type of demand and student satisfaction.

- Attractive demands (A): this type of demands is related to "improvement, environment". They are unanticipated features that surpass student expectations, generating surprise. Even if such demands are not met, students will not feel dissatisfied.

**Table 5. Calculation results of $W_n^{Pos}$ and $W_n^{Neg}$.**

| Student demand | $W_n^{Pos}$ | $W_n^{Neg}$ |
|:---:|:---:|:---:|
| Teacher | 0.0645 | -0.1161 |
| Content | 0.1642 | 0.1697 |
| Structure | -0.0078 | 0.0623 |
| Improvement | 0.1183 | 0.0647 |
| Interaction | 0.0431 | 0.0901 |
| Assessment | 0.1410 | -0.1482 |
| Time | 0.1372 | -0.1524 |
| Difficulty | 0.0791 | 0.1060 |
| Environment | 0.1148 | -0.0276 |
| Material | 0.0077 | 0.0894 |

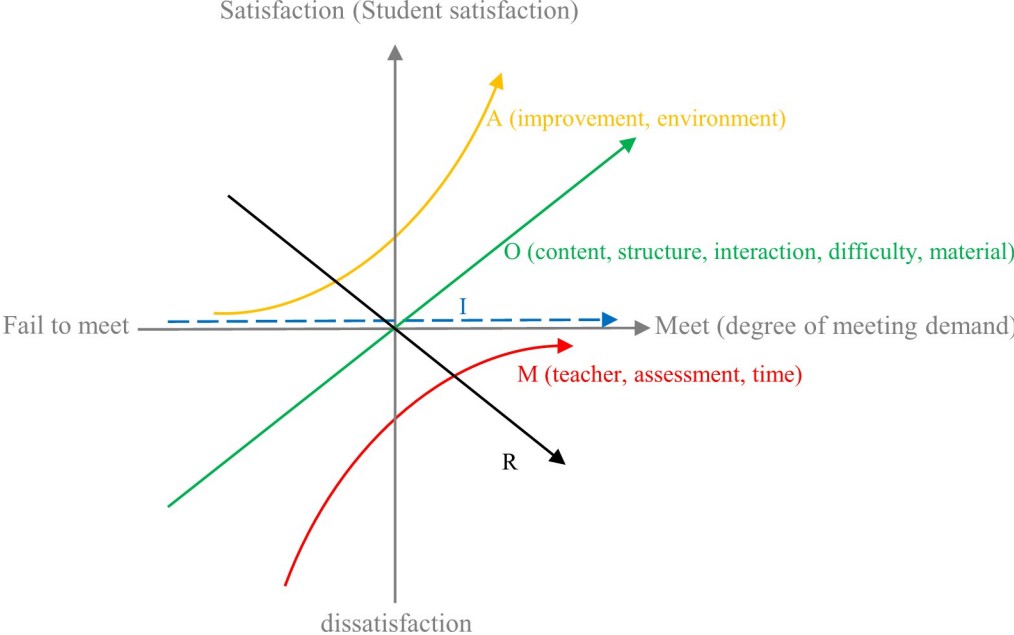

**Fig 9. Classification results for the student demands.**

**(5) MOOC quality improvement strategies.** According to the importance of each demand in Fig 9 (M>O>A), we propose the following strategies to construct high-quality MOOCs in higher vocational education.

First of all, it is necessary to meet the must-be demands of students because they are the most important demands in MOOC learning. In order to construct a group of high-quality MOOCs that cultivate highly qualified and skilled talents, a "double-qualified" teacher team with superb skills and rich theories should be established [36]. At the same time, diversified online assessment methods such as learning tracking (the number of views of MOOC courseware and videos, etc.), paper examinations, and practical reports should be adopted. In addition, it is necessary to fully consider the differences in learning foundation and ability of students in higher vocational institutions, and implement flexible management of academic hours.

Secondly, there is a positive correlation between the degree of meeting students' one-dimensional demands and student satisfaction. Such demands should be met as much as possible under the premise of meeting must-be demands, which will make the MOOCs competitive. The course content should be based on the demands of economic development, support collaborative learning between teachers and students, fully consider the knowledge foundation of students to reasonably determine the difficulty of various courses, and provide students with short videos, audio and other learning resources [37]. In addition, the MOOCs in higher vocational education need to adapt to changing times, and design flexible and modular course structures.

Finally, the MOOCs should strive to meet students' attractive demands. The development of MOOCs in higher vocational education should aim to help students master vocational skills and complete higher education learning. At the same time, it is important to ensure that the MOOC learning platform operates safely, stably and smoothly, in line with the informational level of higher vocational institutions [34].

### 6.3. Practical implications

We have identified and classified students' demands by mining MOOC reviews in higher vocational education to improve MOOC quality. The practical implications of our study are summarized as follows.

First, this study reveals the specific demands of students for teacher quality, course content, interactivity, etc., which provides higher vocational education institutions and teachers with directions for course improvement. For example, in the course Industrial Robot Practical and Application, whose data have been included in our dataset, the teachers should not only be knowledgeable in theory but also have practical experience operating robots in factories. And they should take into full consideration the students' demands for career skills when preparing the lessons and teach the students more skills to operate the robots correctly.

Second, higher vocational education institutions and MOOC platforms can take action under the guidance of our proposed quality improvement strategies to improve students' satisfaction and increase the attractiveness and retention of the courses. For example, higher vocational schools should establish a teacher team with superb skills and rich theories. Teachers should take into full consideration the differences in learning foundation and ability of students in higher vocational institutions when designing course content. MOOC platforms should support diversified learning resources, including audio, short videos, etc., and enhance the platform's interactivity and personalization features to meet the specific demands of students.

## 7. Conclusions and future research

In order to achieve intelligent decision-making for improving the quality of MOOCs in higher vocational education, we propose a data-driven analytics framework for mining and analyzing student reviews in China's higher vocational education MOOCs. Based on web crawlers and text mining, the framework identifies multi-level student demands implicit in MOOC reviews, and classifies the extracted student demands using an artificial neural network and the KANO model. Then based on the above analysis, sustainable MOOC quality improvement strategies are designed.

The data experiments on the website of China University MOOC demonstrate the effectiveness and practicality of the proposed framework. The results show that when designing strategies to improve the quality of the MOOCs in higher vocational education, it is necessary to pay the most attention to students' must-be and one-dimensional demands. And students' attractive demands also need to be met as much as possible.

Our data-driven analytics method provides a unified and extensible framework for studying student demands in the MOOC environment. Our method also provides vocational education practitioners with an intelligent decision-making tool to make sustainable MOOC quality improvement strategies. Future research will consider hypothesis testing on the weights of student demand sentiments, and further explore the time-varying and dynamic characteristics of the student demands.

## Author Contributions

**Conceptualization:** Hongbo Li.

**Funding acquisition:** Hongbo Li.

**Methodology:** Hongbo Li, Huilin Gu.

**Project administration:** Hongbo Li.

**Supervision:** Hongbo Li.

**Validation:** Hongbo Li, Huilin Gu.

**Visualization:** Hongbo Li, Huilin Gu.

**Writing – original draft:** Hongbo Li, Huilin Gu.

**Writing – review & editing:** Hongbo Li, Huilin Gu, Xue Hao, Xin Yan, Qingkang Zhu.

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
