## [Decision Letter · Decision Letter 0]

28 Nov 2023

PONE-D-23-31888Data-driven analytics for student reviews in China’s higher vocational education MOOCs: A quality improvement perspectivePLOS ONE

Dear Dr. Hao,

Thank you for submitting your manuscript to PLOS ONE. After careful consideration, we feel that it has merit but does not fully meet PLOS ONE’s publication criteria as it currently stands. Therefore, we invite you to submit a revised version of the manuscript that addresses the points raised during the review process.

We look forward to receiving your revised manuscript.

Kind regards,

Peida Zhan

Academic Editor

PLOS ONE

 [This research was funded by the Soft Science Project of Shanghai Science and Technology Innovation Action Plan (Grant Number 23692113000)].  

5. We note that Figure(s) 1 and 2 in your submission contain copyrighted images. All PLOS content is published under the Creative Commons Attribution License (CC BY 4.0), which means that the manuscript, images, and Supporting Information files will be freely available online, and any third party is permitted to access, download, copy, distribute, and use these materials in any way, even commercially, with proper attribution. For more information, see our copyright guidelines: http://journals.plos.org/plosone/s/licenses-and-copyright.

a. You may seek permission from the original copyright holder of Figure(s) 1 and 2 to publish the content specifically under the CC BY 4.0 license. 

Additional Editor Comments:

After reading the manuscript, I agree with the suggestions of the two reviewers, and I would like to ask the authors to answer the reviewers' questions one by one and revise the manuscript accordingly. I look forward to your revised manuscript.

Reviewers' comments:

Reviewer's Responses to Questions

**Comments to the Author**

1. Is the manuscript technically sound, and do the data support the conclusions?

Reviewer #1: Yes

Reviewer #2: Yes

2. Has the statistical analysis been performed appropriately and rigorously? 

Reviewer #1: Yes

Reviewer #2: N/A

3. Have the authors made all data underlying the findings in their manuscript fully available?

Reviewer #1: Yes

Reviewer #2: Yes

4. Is the manuscript presented in an intelligible fashion and written in standard English?

Reviewer #1: Yes

Reviewer #2: Yes

5. Review Comments to the Author

Reviewer #1: Thank you for considering PLOS ONE for your submission. In fact, this is a well-written paper. However, there are few comments that need to be addressed to increase the quality of this work.

1. The significance of your study is undoubtedly valuable; however, I suggest that the presentation of its importance could be strengthened. Providing a more explicit and focused outline of the study's significance in the introduction would help readers grasp the broader implications of your research from the outset. Consider clearly articulating the real-world problems or gaps in knowledge that your study addresses, and how your findings contribute to filling those gaps. This will not only engage readers more effectively but also emphasize the practical relevance of your work.

2. While your study touches upon practical implications, there is an opportunity to expand on this aspect further. Readers, especially those from non-academic backgrounds, would benefit from a more detailed exploration of how the findings can be applied in real-world scenarios. Consider dedicating a section to explicitly discussing the practical implications of your research. Address questions such as: How can the results inform decision-making in relevant industries or sectors? Are there specific actions that practitioners can take based on your findings? Providing concrete examples or case studies could illustrate the direct impact of your research on practical applications.

3. Further related literature can be incorporated to enhance the literature review. This includes but not limited to:

- Exploring Student Readiness to MOOCs in Jordan: A Structural Equation Modelling Approach. doi: https://doi.org/10.28945/4542

-

The learning behaviours of dropouts in MOOCs: A collective attention network perspective. doi: https://doi.org/10.1016/j.compedu.2021.104189

Reviewer #2: 1.In lines 48-50, is there an inclusion relationship between "forum data on MOOC websites" and "MOOC reviews"? The specific meaning of the two is suggested to be distinguished.

2.It is suggested to add the enlightenments of existing studies to the literature review.

3.In “Experimental results”, almost every part only provides a visual presentation and brief explanation of the data results, and it is suggested to add more specific and in-depth analysis and interpretation, so as to make the strategies proposed later more reliable.

4.What is the relationship between student demand and student satisfaction? How is the conversion between them in this study?

6. PLOS authors have the option to publish the peer review history of their article (what does this mean?). If published, this will include your full peer review and any attached files.

Reviewer #1: No

Reviewer #2: No

---

## [Author Response · Author response to Decision Letter 0]

17 Dec 2023

Dear Editor,

We first thank you and the reviewers for the helpful and constructive comments. We have thoroughly revised our manuscript according to the suggestions of the reviewers. In our revised manuscript, the revised and newly added text have been marked. Below, please find our point-to-point responses to the comments. For the sake of presentation, the journal requirements and reviewers' comments are duplicated in italics.

Best regards,

The authors (Hongbo Li, Huilin Gu, Xue Hao, Xin Yan, Qingkang Zhu)

Detailed Responses to the academic editor

Response: We have checked the manuscript’s style and revised the formatting irregularities. We have added figure captions following the paragraph in which the figure is first cited.

Response: This research was funded by the Soft Science Project of Shanghai Science and Technology Innovation Action Plan (Grant Number 23692113000). When resubmitting, we will check the ‘Funding Information’ and ‘Financial Disclosure’.

[This research was funded by the Soft Science Project of Shanghai Science and Technology Innovation Action Plan (Grant Number 23692113000)]. 

Response: The funders had no role in study design, data collection and analysis, decision to publish, or preparation of the manuscript. We have amended Role of Funder statement in our cover letter.

Response: We have provided the data. It can be found here: https://github.com/XueHao-23/MOOC-reviews. And we do not make changes to our Data Availability statement.

5. We note that Figure(s) 1 and 2 in your submission contain copyrighted images. All PLOS content is published under the Creative Commons Attribution License (CC BY 4.0), which means that the manuscript, images, and Supporting Information files will be freely available online, and any third party is permitted to access, download, copy, distribute, and use these materials in any way, even commercially, with proper attribution. For more information, see our copyright guidelines: http://journals.plos.org/plosone/s/licenses-and-copyright. 

a. You may seek permission from the original copyright holder of Figure(s) 1 and 2 to publish the content specifically under the CC BY 4.0 license. 

Response: We are unable to obtain permission from the original copyright. So we supply replacement figures for Figures 1 and 2 in the revised manuscript. The screenshots of MOOC websites in Figures 1 and 2 (the original images) are taken to show that the raw data consists of both student reviews and student ratings. Instead of Figure 2 and screenshots in Figure 1, we have plotted a figure showing the composition of the raw data.

Response: We have reviewed our reference list. There are no retracted papers in the reference list. And we have added two papers according to Reviewer 1’s suggestion.

Detailed Responses to Reviewer 1

0. Thank you for considering PLOS ONE for your submission. In fact, this is a well-written paper. However, there are few comments that need to be addressed to increase the quality of this work.

Response: Thanks very much for your comments. We have revised our manuscript and addressed the issues according to your comments. The detailed descriptions of our modifications are given in the following responses.

1. The significance of your study is undoubtedly valuable; however, I suggest that the presentation of its importance could be strengthened. Providing a more explicit and focused outline of the study's significance in the introduction would help readers grasp the broader implications of your research from the outset. Consider clearly articulating the real-world problems or gaps in knowledge that your study addresses, and how your findings contribute to filling those gaps. This will not only engage readers more effectively but also emphasize the practical relevance of your work.

Response: In the revised manuscript, we have explained the significance of this study in more detail in the revised fourth paragraph of the Introduction section. Specifically, first, we summarize the gaps in the existing research. The existing studies not only rarely explore students’ differentiated demands implied in the MOOC reviews but also lack attention to MOOCs in higher vocational education. Then, we propose our research content based on these gaps and clarify our contributions.

2. While your study touches upon practical implications, there is an opportunity to expand on this aspect further. Readers, especially those from non-academic backgrounds, would benefit from a more detailed exploration of how the findings can be applied in real-world scenarios. Consider dedicating a section to explicitly discussing the practical implications of your research. Address questions such as: How can the results inform decision-making in relevant industries or sectors? Are there specific actions that practitioners can take based on your findings? Providing concrete examples or case studies could illustrate the direct impact of your research on practical applications.

Response: In the revised manuscript, we have added a section “6.3 Practical implications” to clarify the practical implications. The practical implications of our study are summarized as follows. 

First, this study reveals the specific demands of students for teacher quality, course content, interactivity, etc., which provides higher vocational education institutions and teachers with directions for course improvement. For example, in the course Industrial Robot Practical and Application, whose data have been included in our dataset, the teachers should not only be knowledgeable in theory but also have practical experience operating robots in factories. And they should take into full consideration the students' demands for career skills when preparing the lessons and teach the students more skills to operate the robots correctly.

Second, higher vocational education institutions and MOOC platforms can take action under the guidance of our proposed quality improvement strategies to improve students' satisfaction and increase the attractiveness and retention of the courses. For example, higher vocational schools should establish a teacher team with superb skills and rich theories. Teachers should take into full consideration the differences in learning foundation and ability of students in higher vocational institutions when designing course content. MOOC platforms should support diversified learning resources, including audio, short videos, etc., and enhance the platform's interactivity and personalization features to meet the specific demands of students.

3. Further related literature can be incorporated to enhance the literature review. This includes but not limited to:

- Exploring Student Readiness to MOOCs in Jordan: A Structural Equation Modelling Approach. doi: https://doi.org/10.28945/4542

-The learning behaviours of dropouts in MOOCs: A collective attention network perspective. doi: https://doi.org/10.1016/j.compedu.2021.104189

Response: In the revised manuscript, we have added and reviewed the above two references in Section 2.1.

Detailed Responses to Reviewer 2

1. In lines 48-50, is there an inclusion relationship between "forum data on MOOC websites" and "MOOC reviews"? The specific meaning of the two is suggested to be distinguished..

Response: In the revised manuscript, we have distinguished the meanings of the two in the introduction. Specifically, forums on MOOC websites support student interaction to learn course content, ask questions, discuss specific topics, and share opinions and experiences, covering a wide range of topics. In this study, MOOC reviews are student evaluations of MOOC resources, which are biased towards students’ experiences of using them. MOOC reviews can be seen as part of forum data on MOOC websites. 

2. It is suggested to add the enlightenments of existing studies to the literature review.

Response: At the end of Section 2, we have summarized the existing studies and analyzed their research gaps to clarify the enlightenment for our research. 

3. In “Experimental results”, almost every part only provides a visual presentation and brief explanation of the data results, and it is suggested to add more specific and in-depth analysis and interpretation, so as to make the strategies proposed later more reliable.

Response: We have complied with this comment. In Section 6.2 of the revised manuscript, we have added more in-depth analysis and interpretation.

4. What is the relationship between student demand and student satisfaction? How is the conversion between them in this study?

Response: In the revised manscript, we have explained the relationship between student demand and student satisfaction in the second paragraph of Section 1, where we first mention student demand and student satisfaction. The degree of meeting student demands affects student satisfaction with MOOCs. In this study, we use the positive and negative sentiment polarity weights of demands to measure the influence of demand on student satisfaction, and use the KANO model to explain the relationship between the degree of meeting various demands and student satisfaction with MOOCs.

---

## [Decision Letter · Decision Letter 1]

30 Jan 2024

Data-driven analytics for student reviews in China’s higher vocational education MOOCs: A quality improvement perspective

PONE-D-23-31888R1

Dear Dr. Hao,

We’re pleased to inform you that your manuscript has been judged scientifically suitable for publication and will be formally accepted for publication once it meets all outstanding technical requirements.

Kind regards,

Peida Zhan

Academic Editor

PLOS ONE

Additional Editor Comments (optional):

After revision, the quality of the article was improved to meet the publication requirements and it is recommended for publication. Congratulations to the authors, and we hope that more in-depth research will be carried out on the basis of this study.

Reviewers' comments:

Reviewer's Responses to Questions

**Comments to the Author**

1. If the authors have adequately addressed your comments raised in a previous round of review and you feel that this manuscript is now acceptable for publication, you may indicate that here to bypass the “Comments to the Author” section, enter your conflict of interest statement in the “Confidential to Editor” section, and submit your "Accept" recommendation.

Reviewer #1: All comments have been addressed

Reviewer #2: All comments have been addressed

2. Is the manuscript technically sound, and do the data support the conclusions?

Reviewer #1: Yes

Reviewer #2: Yes

3. Has the statistical analysis been performed appropriately and rigorously? 

Reviewer #1: Yes

Reviewer #2: Yes

4. Have the authors made all data underlying the findings in their manuscript fully available?

Reviewer #1: No

Reviewer #2: Yes

5. Is the manuscript presented in an intelligible fashion and written in standard English?

Reviewer #1: Yes

Reviewer #2: Yes

6. Review Comments to the Author

Reviewer #1: Thank you for submitting the revised version of your paper. The sharpness and quality of your paper has increased significantly after addressing the reviewers' comments. No further comments.

Reviewer #2: (No Response)

7. PLOS authors have the option to publish the peer review history of their article (what does this mean?). If published, this will include your full peer review and any attached files.

Reviewer #1: No

Reviewer #2: No

---

## [Editor Report · Acceptance letter]

26 Feb 2024

PONE-D-23-31888R1 

PLOS ONE

Dear Dr. Hao, 

I'm pleased to inform you that your manuscript has been deemed suitable for publication in PLOS ONE. Congratulations! Your manuscript is now being handed over to our production team.

Kind regards, 

on behalf of

Dr. Peida Zhan 

Academic Editor

PLOS ONE